# GEOMETRIC INSIGHTS INTO THE CONVERGENCE OF NONLINEAR TD LEARNING

**David Brandfonbrener**
Courant Institute of Mathematical Sciences
New York University
`david.brandfonbrener@nyu.edu`

**Joan Bruna**
Courant Institute of Mathematical Sciences
Center for Data Science
New York University
`bruna@cims.nyu.edu`

## ABSTRACT

While there are convergence guarantees for temporal difference (TD) learning when using linear function approximators, the situation for nonlinear models is far less understood, and divergent examples are known. Here we take a first step towards extending theoretical convergence guarantees to TD learning with nonlinear function approximation. More precisely, we consider the expected learning dynamics of the TD(0) algorithm for value estimation. As the step-size converges to zero, these dynamics are defined by a nonlinear ODE which depends on the geometry of the space of function approximators, the structure of the underlying Markov chain, and their interaction. We find a set of function approximators that includes ReLU networks and has geometry amenable to TD learning regardless of environment, so that the solution performs about as well as linear TD in the worst case. Then, we show how environments that are more reversible induce dynamics that are better for TD learning and prove global convergence to the true value function for well-conditioned function approximators. Finally, we generalize a divergent counterexample to a family of divergent problems to demonstrate how the interaction between approximator and environment can go wrong and to motivate the assumptions needed to prove convergence.

## 1 INTRODUCTION

The instability of reinforcement learning (RL) algorithms is well known, but not well characterized theoretically. Notably, there is no guarantee that value estimation by temporal difference (TD) learning converges when using nonlinear function approximators, even in the on-policy case. The use of a function approximator introduces a projection of the tabular TD update into the class of representable functions. Since the dynamics of TD do not follow the gradient of any objective function, the interaction of the geometry of the function class with that of the TD algorithm in the space of all functions potentially eliminates any convergence guarantees.

This lack of convergence has motivated many authors to seek variants of TD learning that re-establish convergence guarantees, such as two timescale algorithms. In contrast, in this work we focus on TD learning directly and examine its behavior under generic function approximation. We consider the simplest case: on-policy discounted value estimation. To further simplify the analysis, we only consider the expected learning dynamics in continuous time as opposed to the online algorithm with sampling. This means that we are eschewing discussions of off-policy data, exploration, sampling variance, and step size. In this continuous limit, the dynamics of TD learning are modeled as a (nonlinear) ODE. Stability of this ODE is a pre-requisite for convergence of the algorithm. However, for general approximators and MDPs it can diverge as demonstrated by Tsitsiklis & Van Roy (1997). Today, the convergence of this ODE is known in two regimes: under linear function approximation for general environments (Tsitsiklis & Van Roy, 1997) and under reversible environments for general function approximation (Ollivier, 2018). We significantly close this gap through the following contributions:

1. We prove that the set of smooth homogeneous functions, including ReLU networks, is amenable to the expected dynamics of TD. In this case, the ODE is attracted to a compact

set containing the true value function. Moreover, when we use a parametrization inspired by ResNets, nonlinear TD will have error comparable to linear TD in the worst case.

2. We prove global convergence to the true value function when the environment is "more reversible" than the function approximator is "poorly conditioned".

3. We generalize a divergent TD example to a broad class of non-reversible environments.

These results begin to explain how the geometry of nonlinear function approximators and the structure of the environment interact with TD learning.

## 2 SETUP

### 2.1 DERIVING THE DYNAMICS ODE

We consider the problem of on-policy value estimation. Define a Markov reward process (MRP) $\mathcal{M} = (\mathcal{S}, P, r, \gamma)$ where $\mathcal{S}$ is the state space with $|\mathcal{S}| = n$ finite, $P(s'|s)$ is the transition matrix, $r(s, s')$ is the finite reward function, and $\gamma \in [0, 1)$ is the discount factor. This is equivalent to a Markov decision process with a fixed policy. Throughout we assume that $P$ defines an irreducible, aperiodic Markov chain with stationary distribution $\mu$. We want to find the true value function: $V^*(s) := \mathbb{E}[\sum_{t=0}^{\infty} \gamma^t r(s_t, s_{t+1})|s_0 = s]$, where the expectation is taken over transitions from $P$ [1]. The true value function satisfies the Bellman equation:

$$V^*(s) = \mathbb{E}_{s' \sim P(\cdot|s)}[r(s, s') + \gamma V^*(s')]. \tag{1}$$

It will be useful to think of the value function as a vector in $\mathbb{R}^n$ and to define $R(s) = \mathbb{E}_{s' \sim P(\cdot|s)}[r(s, s')]$ so that the Bellman equation becomes $V^* = R + \gamma P V^*$.

The most prominent algorithm for estimating $V^*$ is temporal difference (TD) learning, a form of dynamic programming (Sutton & Barto, 2018). In the tabular setting (with no function approximation) the TD(0) variant of the algorithm with learning rates $\alpha_k$ makes the following update at iteration $k + 1$:

$$V_{k+1}(s) = V_k(s) + \alpha_k(r(s, s') + \gamma V_k(s') - V_k(s)). \tag{2}$$

Under appropriate conditions on the learning rates and noise we have that $V_k \to V^*$ as $k \to \infty$ (Robbins & Monro, 1951; Sutton, 1988). Moreover, under these conditions the algorithm is a discretized and sampled version of the expected continuous dynamics of the following ODE.

$$\dot{V}(s) = \mu(s)(R(s) + \gamma \mathbb{E}_{s' \sim P(\cdot|s)}[V(s')] - V(s)). \tag{3}$$

Letting $D_\mu$ be the matrix with $\mu$ along the diagonal and applying the Bellman equation we get

$$\dot{V} = D_\mu(R + \gamma P V - V) = D_\mu(V^* - \gamma P V^* + \gamma P V - V) = -D_\mu(I - \gamma P)(V - V^*). \tag{4}$$

We now define

$$A := D_\mu(I - \gamma P). \tag{5}$$

While $A$ is not necessarily symmetric, it is positive definite in the sense that $x^T A x > 0$ for nonzero $x$ since $\frac{1}{2}(A + A^T)$ is positive definite (Sutton, 1988). This can be seen by showing that $A$ is a non-singular M-matrix (Horn & Johnson, 1994). This then implies convergence of the ODE defined in (4) to $V^*$ regardless of initial conditions.

In practice, the state space may be too large to use a tabular approach or we may have a feature representation of the states that we think we can use to efficiently generalize. So, we can parametrize a value function $V_\theta$ by $\theta \in \mathbb{R}^d$. Then, the "semi-gradient" TD(0) algorithm is

$$\theta_{k+1} = \theta_k + \alpha_k \nabla V_{\theta_k}(s)(r(s, s') + \gamma V_{\theta_k}(s') - V_{\theta_k}(s)). \tag{6}$$

Now by an abuse of notation, define $V : \mathbb{R}^d \to \mathbb{R}^n$ to be the function that maps parameters $\theta$ to value functions $V(\theta)$ so that $V(\theta)_s = V_\theta(s)$. Now, the associated ODE which we will study becomes:

$$\dot{\theta} = -\nabla V(\theta)^T D_\mu(I - \gamma P)(V(\theta) - V^*) = -\nabla V(\theta)^T A(V(\theta) - V^*). \tag{7}$$

The method is called "semi-gradient" because it is meant to approximate gradient descent on the squared expected Bellman error, which is not feasible since it would require two independent samples of the next state to make each update (Sutton & Barto, 2018). This approximation is what results in the lack of convergence guarantees, as elaborated below.

---

[1] In RL the true value function is often denoted $V^\pi$ for a policy $\pi$. Our MRP can be thought of as an MDP with fixed policy, so $\pi$ is part of the environment and we use $V^*$ to emphasize that the objective is to find $V^*$.

## 2.2 CONVERGENCE FOR LINEAR FUNCTIONS AND REVERSIBLE ENVIRONMENTS

There are two main regimes where the above dynamics are known to converge. The first is when $V(\theta)$ is linear and the second when the MRP is reversible so that $A$ is symmetric.

It is a classic result of Tsitsiklis & Van Roy (1997) that under linear function approximation, where $V(\theta) = \Phi\theta$ for some full rank feature matrix $\Phi$, TD(0) converges to a unique fixed point (the result also applies to the more general TD($\lambda$)). The proof uses the fact that in the linear case, letting $\theta^* = (\Phi^T A\Phi)^{-1}\Phi^T AV^*$, the dynamics (7) become:

$$\dot{\theta} = -\Phi^T A\Phi(\theta - \theta^*).\tag{8}$$

The positive definiteness of $A$ gives global convergence to $\theta^*$.

Recent work has extended this result to give finite sample bounds for linear TD (Bhandari et al., 2018). Making such an extension in the nonlinear case is beyond the scope of this paper since even the simpler-to-analyze expected continuous dynamics are not understood for the nonlinear case.

Some concurrent work has also extended linear convergence to particular kinds of neural networks in the lazy training regime where the networks behave like linear models (Agazzi & Lu, 2019; Cai et al., 2019). The relationship to our work will be discussed in Section 6.

The other main regime where the dynamics converge is when $P$ defines a reversible Markov chain which makes TD(0) gradient descent (Ollivier, 2018). In that case, $A$ is symmetric so (7) becomes:

$$\dot{\theta} = -\frac{1}{2}\nabla\|V(\theta) - V^*\|_A^2\tag{9}$$

where $A$ must be symmetric to define an inner product. Then for any function approximator this gradient flow will approach some local minima. Note that without this symmetry the TD dynamics in (7) are provably not gradient descent of any objective since differentiating the dynamics we get a non-symmetric matrix which cannot be the Hessian of any objective function (Maei, 2011).

## 2.3 AN EXAMPLE OF DIVERGENCE

To better understand the challenge of proving convergence, Tsitsiklis & Van Roy (1997) provide an example of an MRP with zero reward everywhere and a nonlinear function approximator where both the parameters and estimated value function diverge under the expected TD(0) dynamics. We provide a description of this idea in Figure 1.

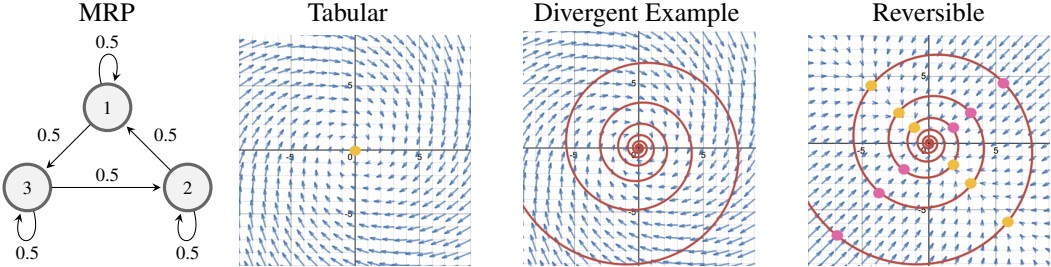

Figure 1: Each dimension of the vector field diagrams corresponds to the value function evaluated at a state. Here we see only a two-dimensional slice of the 3-dimensional function space corresponding to the 3-state MRP. There is no reward so $V^* = 0$. The blue vector field represents the dynamics defined by the linear system $\dot{V} = -A(V - V^*)$. The red spiral represents the one parameter family of functions defined for the divergent counterexample. Using an approximator constrains the dynamics to the red curve by projecting the ambient dynamics (blue arrows) onto the tangent space of the curve (note this projection is not explicitly illustrated in the diagrams). The yellow dots indicate stable fixed points and pink dots unstable fixed points. For the tabular approximator, global convergence to $V^*$ is guaranteed since the dynamics are unconstrained. For the divergent example, projecting the vector field onto the tangent space of the curve causes the dynamics to spiral outwards regardless of initial conditions. However, if we use the same function approximator but make the environment reversible, the dynamics on the curve will converge to a local optimum.

## 3 TD WITH HOMOGENEOUS APPROXIMATORS INCLUDING NEURAL NETWORKS

Our first result is that the expected dynamics of TD(0) are attracted to a neighborhood of the true value function in any irreducible, aperiodic environment when we use a smooth and homogeneous function approximator.

**Definition 1** (Homogeneous). $f : \mathbb{R}^k \to \mathbb{R}^m$ is $h$-homogeneous *for $h \in \mathbb{R}$ if $f(x) = h\nabla f(x)x$. Note that by Euler's homogeneous function theorem, this is equivalent to $f(\alpha x) = \alpha^h f(x)$ for all positive $\alpha$.*

**Remark 1.** *The ReLU activation as well as the square and several others are homogeneous. Moreover, neural networks of any depth with such activations remain homogeneous. This can be found in Lemma 2.1 of (Liang et al., 2017) and we include a proof in Appendix E for completeness.*

Note that linear functions are also homogeneous and we will show that much like linear functions, the set of homogeneous functions works well with TD learning. At a high level, the intuition is that the image of a homogeneous mapping from parameters to functions is a set in function space who's geometry prevents the sort of divergence seen in the spiral example. When $V$ is homogeneous, then the point $V(\theta)$ in the space of functions must lie in the span of the columns of $\nabla V(\theta)$ which define the tangent space to the manifold of functions. This prevents examples like the spiral where the tangent space is nearly orthogonal to $V(\theta)$ for all $V(\theta)$ in the manifold of functions. However, since homogeneous functions are a much more general class than linear functions, the following result is not quite as strong as the global convergence in the linear setting.

**Theorem 1.** *Let $V : \mathbb{R}^d \to \mathbb{R}^n$ be an $h$-homogeneous function such that $\|V(\theta)\|_\mu \leq C\|\theta\|^\ell$. Let $B = \frac{\|(I-\gamma P)V^*\|_\mu}{1-\gamma} = \frac{\|R\|_\mu}{1-\gamma}$. Then, for any initial conditions $\theta_0$, if $\theta$ follows the dynamics defined by (7) we have*

$$\liminf_{t\to\infty} \|V(\theta)\|_\mu \leq B. \tag{10}$$

The full proof is found in Appendix A.1. The main technique is to use homogeneity to see that

$$\frac{d\|\theta\|^2}{dt} = \theta^T \dot\theta = -\theta^T \nabla V(\theta)^T A(V(\theta) - V^*) = -\frac{1}{h}V(\theta)^T A(V(\theta) - V^*). \tag{11}$$

This allows us to relate the norm of the value function to the dynamics in parameter space. We can also extend this result to prove that the limsup of the dynamics attains the same bound if we add a stronger assumption that the approximator is bi-Holder continuous. This result is in Appendix A.2.

One way to think about the theorem is to say that using a homogeneous approximator does at least as well as a baseline given by the zero function. This is because $B$ is a potentially tight bound on $\|V^* - 0\|_\mu$, but cannot be known a priori since we do not know the expected rewards in advance. Using this intuition, we can change the parametrization of the function to include a stronger baseline. We can use a linear baseline since we understand how TD behaves with linear approximators. This gives a parametrization that resembles residual neural networks (He et al., 2016) and which we will call residual-homogeneous when the network is also homogeneous.

**Definition 2** (Residual-homogeneous). *A function $f : \mathbb{R}^{k_1} \times \mathbb{R}^{k_2} \to \mathbb{R}^m$ is residual-homogeneous if $f(x_1, x_2) = \Phi x_1 + g(x_2)$ where $\Phi \in \mathbb{R}^{m \times k_1}$ and $g$ is $h$-homogeneous.*

For this function class, we prove the following theorem, which extend the ideas from Theorem 1.

**Theorem 2.** *Let $V : \mathbb{R}^{d_1} \times \mathbb{R}^{d_2} \to \mathbb{R}^n$ be a residual-homogeneous function where $\Phi$ is a full rank feature matrix and $\|V(\theta)\|_\mu \leq C\|\theta\|^\ell$. Let $\Pi_\Phi$ be the projection onto the span of $\Phi$ and let $B_\Phi = \frac{\|(I-\gamma P)(V^*-\Pi_\Phi V^*)\|_\mu}{1-\gamma}$. Then for any initial conditions $\theta_0$, if $\theta$ follows the dynamics defined by (7) we have*

$$\liminf_{t\to\infty} \|V(\theta) - \Pi_\Phi V^*\|_\mu \leq B_\Phi. \tag{12}$$

The proof is in Appendix A.3. This allows us to bound the quality of the value function found by TD learning as compared to the linear baseline, but we may want to also bound the actual distance to the true value function. Using the above bound relative to the baseline in conjunction with the quality of the baseline, we can derive the following corollary, with proof in Appendix A.3

**Corollary 1.** *Under the same assumptions as Theorem 2 we have*

$$\liminf_{t \to \infty} \|V(\theta) - V^*\|_\mu \leq (1 + \frac{1+\gamma}{1-\gamma})\|V^* - \Pi_\Phi V^*\|_\mu. \tag{13}$$

**Remark 2.** *For linear TD, we get that $\lim_{t \to \infty} \|\Phi\theta - V^*\|_\mu \leq \frac{\|V^* - \Pi_\Phi V^*\|_\mu}{1-\gamma}$ at the fixed point $\theta^*$ (Tsitsiklis & Van Roy, 1997). So, our result shows that in terms of the worst case bound, residual-homogeneous approximators perform similarly to linear functions, especially for large $\gamma$.*

These are the first results that characterize the behavior of TD for a broad class of nonlinear functions including neural networks regardless of initialization under the same assumptions on the environment as used in the analysis of linear TD. Our current results resemble those established in the context of non-convex optimisation using residual networks (Shamir, 2018), also obtained under weak assumptions. One direction for future work would be to extend the results from the liminf to limsup or even to show that the limit exists. Another direction for future work is to try to strengthen the assumptions and leverage structure in $V^*$ itself to reduce the space of possible solutions and be able to make stronger conclusions.

## 4 THE INTERACTION BETWEEN APPROXIMATOR AND ENVIRONMENT

In the previous section we considered a class of approximators for which we can provide guarantees in all irreducible, aperiodic environments. Now we consider how the function approximator interacts with the environment during TD learning. Recall that prior work has shown that in reversible environments, TD learning is performing gradient descent (Ollivier, 2018). Our insight is that strict reversibility is not necessary to make similar guarantees if we also have more information about the function approximator. As seen in the spiral example, the geometric problem with TD arises from the combination of "spinning" linear dynamics from an asymmetric $A$ matrix (i.e. a non-reversible environment) with a poorly conditioned function approximator that "kills" some directions of the update towards the true value function in function space. In this section we will formalize this notion by showing how we can trade off environment reversibility and approximator conditioning and still guarantee convergence. First we need a way to quantify how reversible an environment is and offer the following definition.

**Definition 3** (Reversibility coefficient). *Let $S_A := 1/2(A + A^T)$ and $R_A := 1/2(A - A^T)$ be the symmetric and anti-symmetric parts of A, as defined in equation (5). Then the reversibility coefficient $\rho(\mathcal{M})$ is*

$$\rho(\mathcal{M}) := \min_{V \in \mathbb{R}^n \setminus \{0\}} \frac{\|S_A V\|^2 + \|AV\|^2}{\|R_A V\|^2} \tag{14}$$

Note that when the environment is reversible this coefficient is infinite since in that case $A$ is symmetric and $R_A$ is zero. The antisymmetric part $R_A$ captures the spinning behavior of the linear dynamical system in function space (as in the spiral example). At a high level, more spinning means a less reversible environment and larger $R_A$ which lowers the reversibility coefficient.

Now we need a compatible way to quantify the effect of the function approximator. To do this, we have to examine the matrix $\nabla V(\theta) \nabla V(\theta)^T$. This matrix shapes the dynamics of TD and in the case of neural networks under particular assumptions it is known as the neural tangent kernel (Jacot et al., 2018). The condition number of this matrix gives us one way to quantify how much the approximator prefers updates along the directions in function space corresponding to maximal eigenvalues over those corresponding to minimal eigenvalues. This gives us a way to quantify how well-behaved the function approximator is and allows us to prove the following theorem.

**Theorem 3.** *Let $\kappa(M)$ be the condition number of a matrix $M$. Assume that for all $\theta$,*

$$\kappa(\nabla V(\theta) \nabla V(\theta)^T) < \rho(\mathcal{M}). \tag{15}$$

*Then if $\theta$ evolves according to (7) we have that for all $\theta$*

$$\frac{d\|V(\theta) - V^*\|_{S_A}^2}{dt} < 0 \tag{16}$$

*where $S_A := 1/2(A + A^T)$. Thus, $V(\theta) \to V^*$ regardless of initial conditions.*

The proof is relatively simple and proceeds by using the chain rule to write out the time derivative of the Lyapunov function and applying the Courant-Fischer-Weyl min-max theorem along with the assumption to get the result. The full details can be found in Appendix B.

This result provides strong global convergence guarantees, albeit under fairly strong assumptions. It nevertheless provides intuition about how the environment interacts with the function approximation. We show that we can use a nonlinear approximator that generalizes across states by using gradient information so long as the condition number of the tangent kernel of the approximator is bounded by the reversibility coefficient. Note that for the condition number of the kernel to be finite, the Jacobian must be full rank which means that the function approximator has more parameters than the number of states. Such an approximator has more parameters than a tabular approximator, but can be nonlinear and generalize across states using the structure of the input representation. This opens a few directions for future work to formalize the relationship between the environment and approximator in the regime when there are less parameters than states or when the state space is infinite. It may be fruitful to connect this to the literature from supervised learning on over-parametrized neural networks (see for example Oymak & Soltanolkotabi (2019) and references therein), especially in the case of value estimation from a finite dataset (i.e. a replay buffer). Another direction would be to leverage structure in $V^*$ and the input representation so that the approximator is effectively over-parametrized and similar arguments can be made, but it is not clear how to formalize such assumptions.

### 4.1 EXTENSION TO K-STEP RETURNS

We now consider how the analysis strategy presented above applies to a classical variation on the TD learning algorithm. We find that $k$-step returns have better convergence guarantees by increasing the reversibility of the effective environment. This is not completely surprising since in the limit $k \to \infty$ we recover gradient descent in the $\mu$-norm with the Monte Carlo algorithm for value estimation. However, we show that our sufficient condition to guarantee convergence in the well-conditioned regime weakens exponentially with $k$. In Appendix C we show that with $k$ step returns the dynamics of the algorithm become

$$\dot{\theta} = -\nabla V(\theta)^T D_\mu (I - (\gamma P)^k)(V(\theta) - V^*). \tag{17}$$

We can define a notion of effective reversibility that scales exponentially with $k$ such that we recover the same type of convergence as Theorem 3 whenever

$$\kappa(\nabla V(\theta) \nabla V(\theta)^T)^{1/2} < \frac{\mu_{min}(1 - \gamma^k)}{\mu_{max}}(\gamma \lambda_2(P))^{-k} \tag{18}$$

where $\lambda_2(P)$ is the second largest eigenvalue of $P$, which is strictly less than 1 under our irreducible and aperiodic assumption. This result shows how using $k$-step returns to increase the effective reversibility of the environment can lead to better convergence properties. See Appendix C for a more precise statement of the result and its proof.

### 4.2 NUMERICAL EXPERIMENT ON THE DIVERGENT EXAMPLE

We perform a small set of experiments on the divergent spiral example from Section 2.3 which support our conclusions about reversibility and $k$-step returns. We integrate the expected dynamics ODEs in two settings, one where we introduce reversibility into the environment and the other where we increase the value of $k$ in the algorithm. The function approximator is always the spiral approximator from the example. We can introduce reversibility by adding reverse connections to the environment with probability $\delta \in \{0, 0.1, 0.2, 0.23\}$ as shown in Figure 2. This effectively reduces the spinning of the linear dynamical system in function space defined by the Bellman operator. We find that increasing reversibility eventually leads to convergence. We also validate the result that increasing $k$ will lead to convergence by increasing the effective reversibility without changing the MRP. Note that the spiral example is outside the assumptions of the theory in this section since the function is not well-conditioned, but we wanted to show that the connection between reversibility and convergence may extend beyond the well-conditioned setting.

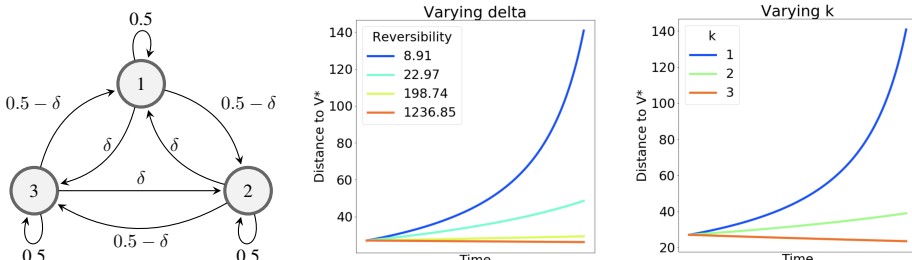

Figure 2: Left: the Markov chain used for the experiments labeled with the transition probabilities. Center: the spiral divergence example in progressively more reversible environments. A stronger reverse connection makes the environment more reversible and eventually causes convergence. Right: The impact of using $k$-step returns. We use $\delta = 0$ for all values of $k$ and get convergence for $k = 3$.

## 5 A GENERALIZED DIVERGENT EXAMPLE

To motivate the necessity of assumptions similar to the ones that we have made we can look again to the spiral example of (Tsitsiklis & Van Roy, 1997). Here we generalize this example to arbitrary number of states for most non-reversible MRPs. Our construction allows for approximators with arbitrary number of parameters, but restricts them to have rank deficient tangent kernels to mimic the spiral in a 2-D subspace of function space. The construction can be found in Appendix D, and the result can be described formally as follows.

**Proposition 1.** *If the MRP is not reversible such that $A = D_\mu(I - \gamma P)$ has at least one non-real eigenvalue, then there exists a function approximator $V$ such that TD learning will diverge. That is, for any initial parameters $\theta_0$, as $t \to \infty$ we have $\|V(\theta) - V^*\| \to \infty$. Moreover, $\nabla V(\theta)$ can have rank up to $n - 1$, where $n$ is the number of states, for all $\theta$.*

The construction of the approximator in the counterexample is somewhat pathological, but any convergence proofs have to make assumptions to rule out these divergent examples. In this work we avoid these by using either smooth homogeneous functions or by using well-conditioned functions in nearly-reversible environments. While there may be other assumptions that yield convergence, they must also account for this class of divergent examples.

## 6 RELATED WORK

### 6.1 CONNECTIONS TO WORK IN THE LAZY TRAINING REGIME

Concurrent work (Agazzi & Lu, 2019) has proven convergence of expected TD in the nonlinear, non-reversible setting in the so-called "lazy training" regime, in which nonlinear models (including neural networks) with particular parametrization and scaling behave as linear models, with a kernel given by the linear approximation of the function at initialization. Whereas this kernel captures some structure from the function approximation, the lazy training regime does not account for feature selection, since parameters are confined in a small neighborhood around their initialization (Chizat & Bach, 2018). Another result in a similar direction is from concurrent work (Cai et al., 2019) which considers two-layer networks (one hidden layer) in the large width regime where only the first layer is trained. They show that this particular type of function with fixed output layer is nearly linear and derive global convergence in the limit of large width with an additional assumption on the regularity of the stationary distribution. In contrast with these works, our results account for feature selection with more general nonlinear functions. Our homogeneous results hold for a broad class of approximators much closer to those used in practice and our well-conditioned results hold for general nonlinear parametrization and provide useful intuition about the relationship between approximator and environment.

## 6.2 CONNECTIONS TO WORK ON FITTED VALUE ITERATION

Another line of work provides convergence rates for fitted value iteration or fitted Q iteration under the assumption of small optimization error at each iteration (Munos, 2007; Munos & Szepesvári, 2008; Yang et al., 2019b). These papers give bounds that depend on the maximum difference between the function returned by the Bellman operator applied to the current iterate and its projection into the space of representable functions (which they call the inherent Bellman residual). This assumption means that a priori the function class has geometry amenable to the MDP being evaluated. Here we do not rely on any assumptions about successful optimization or an assumption that the projection of the tabular TD update into the space of representable functions is uniformly small. Instead we find scenarios where we can guarantee that the difference between the tabular update and projection cannot be too far so that the optimization procedure succeeds.

## 6.3 ALTERNATIVE VALUE ESTIMATION ALGORITHMS

There are several papers that introduce new algorithms inspired by TD learning but modified so as to have provable convergence with nonlinear approximators. To our knowledge, all of them use either a two timescale argument where the optimization procedure at the faster timescale views the slower timescale as fixed (Borkar, 1997; 2008) or they attempt to optimize a different objective function (Baird, 1995). Most of the algorithms have not seen widespread use, potentially because these modifications make optimization more difficult or decrease the quality of solutions. More specifically, Baird (1995) present residual algorithms, which attempt to optimize a different objective which avoids double sampling, but has incorrect value functions as solutions (Sutton & Barto, 2018). Bhatnagar et al. (2009) present GTD2/TDC which uses two timescales to perform gradient descent on the norm of the TD(0) dynamics projected onto the image of the nonlinear approximator in function space and thus has the same fixed points as TD(0). More recently, Dai et al. (2018) and Chung et al. (2019) present two timescale algorithms which are provably convergent. Yang et al. (2019a) characterize target networks, which have seen widespread use (Mnih et al., 2015), as a two timescale algorithm. Finally, while they do not provide guaranteed convergence with nonlinear functions, Achiam et al. (2019) present an algorithm that uses a similar observation to ours about the connection between TD-learning with nonlinear functions and the neural tangent kernel. Their algorithm then estimates a preconditioning matrix that serves a similar function as the two-timescale argument. In contrast to this line of work on algorithmic modifications, our work is a first step towards characterizing the behavior of nonlinear TD without two timescales or a modified objective.

## 6.4 EMPIRICAL WORK

Recent empirical work by Fu et al. (2019) empirically investigates the interaction between nonlinear function approximators (neural networks) with Q-learning, which is a semi-gradient algorithm that is similar to TD. They find that divergence is rare and that more expressive approximators reduce approximation error and reduce the chances of divergence. Since these more expressive approximators are more likely to be well-conditioned, this gives reason to believe it may be possible to extend our convergence results from the well-conditioned setting for very expressive approximators to more realistic approximators.

## 7 DISCUSSION

We have considered the expected continuous dynamics of the TD algorithm for on policy value estimation from the perspective of the interaction of the geometry of the function approximator and environment. Using this perspective we derived two positive results and one negative result. First, we showed attraction to a compact set when homogeneous approximators like ReLU networks. The worst case solution in this set is comparable to the worst case linear TD solution for a particular parametrization inspired by ResNets. Second, we showed global convergence when the environment is more reversible than the approximator is poorly-conditioned. Finally, we provided a generalized counterexample to motivate the assumptions necessary to rule out bad interactions between approximator and environment.

There are several possible directions for future work. First, while our results extend both the linear and reversible convergence regimes, they do not close the gap between the two. One direction for

future work is thus to provide a unifying analysis that would neatly connect all of the convergent regimes. Next, it may be possible to find a more precise notion of the well-conditioning necessary to get local convergence rather than global convergence which would allow extension to more realistic settings. It may be possible to leverage assumptions about regularity of the true value function so that the function class is effectively well-conditioned. Another direction is that, while it was beyond the scope of this paper, it would be instructive to extend the results to finite sample results as has recently been done for linear TD. It would also be interesting to extend the results to off-policy and Q-learning settings, but likely would require stronger assumptions. Finally, we would like to motivate future work by noting that the ultimate goal of this line of work is to put TD on the same solid footing as optimization in supervised learning where we can characterize easy problems (by convexity), can guarantee convergence even in hard problems (to local minima), and have some notions of how to make optimization easier (like over-parametrization). Here we have taken a step towards this kind of analysis, but the precise characterization of what makes a problem easy and whether and where TD converges on hard problems remain incomplete.

### ACKNOWLEDGMENTS

We would like to thank Yann Ollivier for helping to inspire this project and for sharing some of his notes with us. We also thank the lab mates, especially Will Whitney, Aaron Zweig, and Min Jae Song, who provided useful discussions and feedback.

This work was partially supported by the Alfred P. Sloan Foundation, NSF RI-1816753, NSF CA-REER CIF 1845360, and Samsung Electronics.

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

## A    TD with homogeneous approximators

### A.1    Proof of Theorem 1

**Theorem 1.** *Let $V : \mathbb{R}^d \to \mathbb{R}^n$ be an h-homogeneous function such that $\|V(\theta)\|_\mu \leq C\|\theta\|^\ell$. Let $B = \frac{\|(I-\gamma P)V^*\|_\mu}{1-\gamma} = \frac{\|R\|_\mu}{1-\gamma}$. Then, for any initial conditions $\theta_0$, if $\theta$ follows the dynamics defined by (7) we have*

$$\liminf_{t \to \infty} \|V(\theta)\|_\mu \leq B. \tag{10}$$

*Proof.* Applying the chain rule, (7), and the homogeneity assumption we get that:

$$\frac{d\|\theta\|^2}{dt} = \theta^T \dot{\theta} = -\theta^T \nabla V(\theta)^T A(V(\theta) - V^*) = -\frac{1}{h}V(\theta)^T A(V(\theta) - V^*).$$

For any $\epsilon > 0$, whenever $B + \epsilon = \frac{\|(I-\gamma P)V^*\|_\mu}{1-\gamma} + \epsilon < \|V(\theta)\|_\mu$ we have that

$$\begin{aligned}
|V(\theta)^T AV^*| &\leq \|V(\theta)\|_\mu \|(I-\gamma P)V^*\|\mu \\
&< (1-\gamma)\|V(\theta)\|_\mu^2 - \epsilon(1-\gamma)\|V(\theta)\|_\mu \\
&\leq \|V(\theta)\|_\mu^2 - \gamma\|V(\theta)\|_\mu^2 - \epsilon(1-\gamma)(B+\epsilon) \\
&\leq V(\theta)^T D_\mu V(\theta) - \gamma V(\theta)^T D_\mu PV(\theta) - c \\
&= V(\theta)^T AV(\theta) - c
\end{aligned}$$

for $c = \epsilon(1-\gamma)(B+\epsilon) > 0$. The last inequality follows from an application of Lemma 1 of (Tsitsiklis & Van Roy, 1997) that $\|PV\|_\mu \leq \|V\|_\mu$ along with Cauchy-Schwarz to show that:

$$V(\theta)^T D_\mu PV(\theta) = V(\theta)^T D_\mu^{1/2} D_\mu^{1/2} PV(\theta) \leq \|V(\theta)\|_\mu \|PV(\theta)\|_\mu \leq \|V(\theta)\|_\mu^2.$$

Putting this all together we get that whenever $B + \epsilon < \|V(\theta)\|_\mu$ we have that

$$\frac{d\|\theta\|^2}{dt} \leq -\frac{1}{h}(V(\theta)^T AV(\theta) - |V(\theta)^T AV^*|) < -\frac{c}{h} < 0.$$

Put another way, we can define $U = \{\theta : \frac{d\|\theta\|^2}{dt} \geq -c/h\}$ and $W = \{V : \|V\|_\mu \leq B + \epsilon\}$. By the above, we have that $V(U) \subseteq W$ where $V(U)$ is the image of $U$ under $V$, i.e. $\{V : \exists \theta \in U \; s.t. \; V = V(\theta)\}$. Now define $\mathcal{O} = \{\theta : C\|\theta\|^\ell < B + \epsilon\}$ which contains an open ball around the origin. By our Holder continuity assumption, we know that $V(\mathcal{O}) \subseteq W$.

Putting it all together, if $\theta \notin U$ at time $t$, the dynamics of $\|\theta\|^2$ are contracting faster than $c/h$ so that there exists a finite $T > t$ such that at time $T$ the parameters $\theta(T)$ will either be in $\mathcal{O}$ or $U$, either way implying $V(\theta)$ will be in $W$. Since the above reasoning held for any $\epsilon > 0$, taking the liminf we can take $\epsilon \to 0$ which yields the result. $\square$

### A.2    Bi-Holder homogeneous stability

**Proposition 2.** *Let $V : \mathbb{R}^d \to \mathbb{R}^n$ be h-homogeneous. Moreover, assume that $c\|\theta\|^s \leq \|V(\theta)\|_\mu \leq C\|\theta\|^r$ for some $c, s, r, C > 0$. Let $B = \frac{\|(I-\gamma P)V^*\|_\mu}{1-\gamma} = \frac{\|R\|_\mu}{1-\gamma}$. Then, for any initial conditions $\theta_0$, if $\theta$ follows the dynamics defined by (7) we have*

$$\limsup_{t \to \infty} \|V(\theta)\|_\mu \leq C(B/c)^{r/s}. \tag{19}$$

*Proof.* Using the same reasoning as above without the additional $\epsilon$, we get that whenever $B = \frac{\|(I-\gamma P)V^*\|_\mu}{1-\gamma} < \|V(\theta)\|_\mu$, we have

$$|V(\theta)^T AV^*| \leq V(\theta)^T AV(\theta).$$

So, whenever $B < \|V(\theta)\|_\mu$, we have that $\frac{d\|\theta\|^2}{dt} < 0$.

Define $\mathcal{O} = \{\theta : c\|\theta\|^s \leq B\}$. Thus, by the lower Holder bound, $\frac{d\|\theta\|^2}{dt} < 0$ for all $\theta \notin \mathcal{O}$. So, we have that as $t \to \infty$, $\theta \in \mathcal{O}$. And, by the upper Holder bound, if $\theta \in \mathcal{O}$ then

$$\|V(\theta)\|_\mu \leq \max_{\theta' \in \mathcal{O}} C\|\theta'\|^r \leq C(B/c)^{r/s}$$

Since, $\theta \in \mathcal{O}$ in the limit, we get the desired result. $\qquad\square$

### A.3 PROOF OF THEOREM 2 AND COROLLARY 1

**Theorem 2.** *Let* $V : \mathbb{R}^{d_1} \times \mathbb{R}^{d_2} \to \mathbb{R}^n$ *be a residual-homogeneous function where* $\Phi$ *is a full rank feature matrix and* $\|V(\theta)\|_\mu \leq C\|\theta\|^\ell$. *Let* $\Pi_\Phi$ *be the projection onto the span of* $\Phi$ *and let* $B_\Phi = \frac{\|(I - \gamma P)(V^* - \Pi_\Phi V^*)\|_\mu}{1 - \gamma}$. *Then for any initial conditions* $\theta_0$, *if* $\theta$ *follows the dynamics defined by (7) we have*

$$\liminf_{t \to \infty} \|V(\theta) - \Pi_\Phi V^*\|_\mu \leq B_\Phi. \tag{12}$$

*Proof.* Let $f$ be homogeneous and

$$V(\theta) = V(\theta_1, \theta_2) = \Phi\theta_1 + f(\theta_2)$$

Let $\theta_1^* = \Pi_\Phi V^*$ be the best linear predictor. In fact, the proof would go through for any $\theta_1^*$, only the bound gets worse for worse choice of $\theta_1^*$.

Note that

$$\begin{aligned}
\frac{d\|\theta - (\theta_1^*, 0)\|^2}{dt} &= -(\theta - (\theta_1^*, 0))^\top DV(\theta)^\top A(V(\theta) - V^*) \\
&= -(\Phi(\theta_1 - \theta_1^*) + f(\theta_2 - 0))^\top A(V(\theta) - V^*) \\
&= -(V(\theta) - \Phi\theta_1^*)^\top A(V(\theta) - \Phi\theta_1^*) - (V(\theta) - \Phi\theta_1^*)^\top A(\Phi\theta_1^* - V^*).
\end{aligned}$$

The residual parametrization is necessary here to have the linear part of $DV(\theta)$ independent from $\theta$ so that we can turn the difference of $\theta_1 - \theta_1^*$ into a difference between functions.

As in the proof of Theorem 1, for any $\epsilon > 0$ we get that whenever $B_\Phi + \epsilon < \|V(\theta) - \Phi\theta_1^*\|_\mu^2$,

$$\begin{aligned}
|(V(\theta) - \Phi\theta_1^*)^T A(\Phi\theta_1^* - V^*)| &\leq \|V(\theta) - \Phi\theta_1^*\|_\mu \|(I - \gamma P)(\Phi\theta_1^* - V^*)\|_\mu \\
&< (1 - \gamma)\|V(\theta) - \Phi\theta_1^*\|_\mu^2 - \epsilon(1 - \gamma)\|V(\theta) - \Phi\theta_1^*\|_\mu^2 \\
&\leq \|V(\theta) - \Phi\theta_1^*\|_\mu^2 - \gamma\|V(\theta) - \Phi\theta_1^*\|_\mu^2 - \epsilon(1 - \gamma)(B_\Phi + \epsilon) \\
&\leq \|V(\theta) - \Phi\theta_1^*\|_\mu^2 - \gamma(V(\theta) - \Phi\theta_1^*)^T D_\mu P(V(\theta) - \Phi\theta_1^*) - c \\
&= (V(\theta) - \Phi\theta_1^*)^T A(V(\theta) - \Phi\theta_1^*) - c.
\end{aligned}$$

As a consequence we can conclude that whenever $B_\Phi + \epsilon < \|V(\theta) - \Phi\theta_1^*\|_\mu^2$

$$\frac{d\|\theta - (\theta_1^*, 0)\|^2}{dt} < -c/h.$$

To conclude the proof, we can define $U = \{\theta : \frac{d\|\theta - (\theta_1^*, 0)\|^2}{dt} \geq -c/h\}$ and $W = \{V : \|V - \Phi\theta_1^*\|_\mu \leq B + \epsilon\}$. Then we have that $V(U) \subseteq W$. Now define $\mathcal{O} = \{(\theta_1^*, 0) + \theta : C\|\theta\|^\ell < B + \epsilon\}$ which contains an open ball around $(\theta_1^*, 0)$. By our Holder continuity assumption and the residual parametrization of $V$, we know that $V(\mathcal{O}) \subseteq W$.

Putting it all together, if $\theta \notin U$ at time $t$, the dynamics of $\|\theta\|^2$ are contracting faster than $c/h$ so that there exists a finite $T > t$ such that at time $T$ the parameters $\theta(T)$ will either be in $\mathcal{O}$ or $U$, either way implying $V(\theta)$ will be in $W$. Since the above reasoning held for any $\epsilon > 0$, taking the liminf we can take $\epsilon \to 0$ which yields the result. $\qquad\square$

**Corollary 1.** *Under the same assumptions as Theorem 2 we have*

$$\liminf_{t \to \infty} \|V(\theta) - V^*\|_\mu \leq (1 + \frac{1 + \gamma}{1 - \gamma})\|V^* - \Pi_\Phi V^*\|_\mu. \tag{13}$$

*Proof.* Applying the triangle inequality and the result of the Theorem, we get that

$$\liminf_{t\to\infty} \|V(\theta) - V^*\|_\mu \leq \liminf_{t\to\infty}(\|V(\theta) - \Pi_\Phi V^*\|_\mu + \|\Pi_\Phi V^* - V^*\|_\mu)$$

$$= \liminf_{t\to\infty} \|V(\theta) - \Pi_\Phi V^*\|_\mu + \|\Pi_\Phi V^* - V^*\|_\mu$$

$$\leq B_\Phi + \|\Pi_\Phi V^* - V^*\|_\mu$$

Then, we note that $\|I - \gamma P\| \leq 1 + \gamma$ since $P$ is a stochastic matrix which lets us bound $B_\Phi$ by $\frac{1+\gamma}{1-\gamma}\|\Pi_\Phi V^* - V^*\|_\mu$. Putting this together with the above yields the result. $\square$

# B CONVERGENCE IN THE WELL-CONDITIONED SETTING

## B.1 PROOF OF THE THEOREM

**Theorem 3.** *Let $\kappa(M)$ be the condition number of a matrix $M$. Assume that for all $\theta$,*

$$\kappa(\nabla V(\theta)\nabla V(\theta)^T) < \rho(\mathcal{M}). \tag{15}$$

*Then if $\theta$ evolves according to (7) we have that for all $\theta$*

$$\frac{d\|V(\theta) - V^*\|_{S_A}^2}{dt} < 0 \tag{16}$$

*where $S_A := 1/2(A + A^T)$. Thus, $V(\theta) \to V^*$ regardless of initial conditions.*

*Proof.* To simplify notation, we use $S$ for $S_A$ and $R$ for $R_A$. We define $L : \mathbb{R}^d \to \mathbb{R}$ which will be the Lyapunov function for the dynamical system as

$$L(\theta) = \|V(\theta) - V^*\|_S^2$$

By applying the chain rule and the polarization identity to (7) we have

$$\dot{L}(\theta) = -\left\langle \nabla V(\theta)^T(A + A^T)(V(\theta) - V^*), \nabla V(\theta)^T A(V(\theta) - V^*) \right\rangle$$

$$= -\|\nabla V(\theta)^T A(V(\theta) - V^*)\|^2 - \left\langle \nabla V(\theta)^T A^T(V(\theta) - V^*), \nabla V(\theta)^T A(V(\theta) - V^*) \right\rangle$$

$$= -\|\nabla V(\theta)^T A(V(\theta) - V^*)\|^2$$

$$\quad - \frac{1}{4}\left( \|\nabla V(\theta)^T(A + A^T)(V(\theta) - V^*)\|^2 - \|\nabla V(\theta)^T(A - A^T)(V(\theta) - V^*)\|^2 \right)$$

$$= -\|\nabla V(\theta)^T A(V(\theta) - V^*)\|^2 - \|\nabla V(\theta)^T S(V(\theta) - V^*)\|^2 + \|\nabla V(\theta)^T R(V(\theta) - V^*)\|^2$$

where $R := 1/2(A - A^T)$. So using the assumption and then the min-max theorem,

$$\kappa(\nabla V(\theta)\nabla V(\theta)^T) < \rho(\mathcal{M})$$

$$\lambda_{\max}(\nabla V(\theta)\nabla V(\theta)^T)\|R(V(\theta) - V^*)\|^2 < \lambda_{\min}(\nabla V(\theta)\nabla V(\theta)^T)\left( \|A(V(\theta) - V^*)\|^2 + \|S(V(\theta) - V^*)\|^2 \right)$$

$$\|\nabla V(\theta)^T R(V(\theta) - V^*)\|^2 < \|\nabla V(\theta)^T A(V(\theta) - V^*)\|^2 + \|\nabla V(\theta)^T S(V(\theta) - V^*)\|^2$$

where $\lambda_{\max}, \lambda_{\min}$ are the maximal and minimal eigenvalues. Combining, we conclude that $\dot{L}(\theta) < 0$ and the result follows. $\square$

## B.2 CALCULATING THE REVERSIBILITY COEFFICIENT

The reversibility coefficient can be seen as the minimizer of a generalized Rayleigh quotient (Horn & Johnson, 1994). Such a quotient for Hermitian matrices $B, C$ is defined as $R_{B,C}(u) = \frac{u^T B u}{u^T C u}$. And when $C$ is full rank, we know that this is maximized by the maximal eigenvalue of $C^{-1}B$. In our case, this gives us a way to calculate the reversibility coefficient as

$$\rho(\mathcal{M}) = \left( \max_{V \in \mathbb{R}^n \setminus \{0\}} \frac{\|R_A V\|^2}{\|S_A V\|^2 + \|AV\|^2} \right)^{-1} = \left( \lambda_{\max}([S_A^T S_A + A^T A]^{-1} R_A^T R_A) \right)^{-1}. \tag{20}$$

## C   K-STEP RETURNS IN THE WELL-CONDITIONED SETTING

### C.1   EXPECTED DYNAMICS DERIVATION

The $k$-step return variant of TD learning replaces $r(s, s') + \gamma V_\theta(s')$ in the original algorithm (6) by $\sum_{t=0}^{k-1} \gamma^t r(s_t, s_{t+1}) + \gamma^k V_\theta(s_k)$. To get the continuous matrix version of the dynamics we replace $R$ with $V^* - \gamma P V^*$ in the following expression to get a telescoping series so that

$$V - \sum_{t=0}^{k-1} (\gamma P)^t R + (\gamma P)^k V = V - V^* - (\gamma P)^k (V - V^*) = (I - (\gamma P)^k)(V - V^*).$$

Thus, we can define the TD($k$) dynamics by

$$\dot{\theta} = -\nabla V(\theta)^T D_\mu (I - (\gamma P)^k)(V(\theta) - V^*). \tag{21}$$

### C.2   EFFECTIVE REVERSIBILITY

To draw the comparison to the TD(0) algorithm analyzed in the paper, we define the matrix

$$A_k := D_\mu (I - (\gamma P)^k).$$

Now we take symmetric and asymmetric parts $S_k = \frac{1}{2}(A_k + A_k^T)$ and $R_k = \frac{1}{2}(A_k - A_k^T)$. Then we can define reversibility in this effective environment.

**Definition 4** (Effective reversibility coefficient).

$$\rho_k(\mathcal{M}) := \min_{V \in \mathbb{R}^n \setminus \{0\}} \frac{\|S_k V\|^2 + \|A_k V\|^2}{\|R_k V\|^2}. \tag{22}$$

### C.3   CONVERGENCE PROPOSITION

**Proposition 3.** *Let $\lambda_2(P)$ be the modulus of the second largest eigenvalue of $P$, which is strictly less than 1 by our irreducible, aperiodic assumption. Then*

$$\frac{\mu_{min}(1 - \gamma^k)}{\mu_{max}}(\gamma \lambda_2(P))^{-k} \leq \rho_k(\mathcal{M})^{1/2} \tag{23}$$

*And if $\theta$ follows the ODE defined by (21), then $\frac{d\|V(\theta) - V^*\|_{S_k}^2}{dt} < 0$ whenever*

$$\kappa(\nabla V(\theta) \nabla V(\theta)^T)^{1/2} < \frac{\mu_{min}(1 - \gamma^k)}{\mu_{max}}(\gamma \lambda_2(P))^{-k}. \tag{24}$$

*As a consequence, as long as $\kappa(\nabla V(\theta) \nabla V(\theta)^T)$ is finite there exists $k$ sufficiently large to guarantee convergence to $V^*$ regardless of initial conditions.*

*Proof.* To extend Theorem 3 to this result, we will show that

$$\frac{\mu_{min}(1 - \gamma^k)}{\mu_{max}}(\gamma \lambda_2(P))^{-k} < \min_{V \in \mathbb{R}^n \setminus \{0\}} \frac{\|S_k V\|}{\|R_k V\|} \leq \rho_k(\mathcal{M})^{1/2}$$

First, since $R_k = \frac{1}{2}((\gamma P^T)^k D_\mu - D_\mu(\gamma P)^k)$ we have $R_k \mathbf{1} = 0$ since $D_\mu$ is the stationary distribution so that $(P^T)^k D_\mu \mathbf{1} = D_\mu P^k \mathbf{1} = \mu$. So we only need to consider $V \perp \mathbf{1}$. Now, we have

$$\|R_k V\| \leq \frac{\gamma^k}{2}\left(\|(P^T)^k D_\mu V\| + \|D_\mu P^k V\|\right) \leq \mu_{max} \gamma^k (\lambda_2(P))^k \|V\|$$

where $\lambda_2$ is the eigenvalue with second largest magnitude (since 1 is the largest eigenvalue, but it has eigenvector $\mathbf{1}$ for P and $\mu$ for $P^T$). And we have

$$\|S_k V\| \geq \lambda_{min}(S_k)\|V\| \geq (\mu_{min} - \gamma^k \mu_{min})\|V\|$$

by the Gershgorin circle theorem since $P^k$ defines a distribution with stationary distribution $D_\mu$ so that subtracting the off diagonal from the diagonal of $A_k$ we get $\lambda_{min}(A_k) \geq \mu_i - \sum_j \gamma^k D_\mu P_{ij}^k =$

$\mu_i - \gamma^k \mu_i$ for all $i$. Since an analogous argument applies to the columns of $A_k$, we get the above bound on the eigenvalues of $S_k$. Dividing the bounds we get the first result.

Then by Theorem 3, we get the second result that $\frac{d\|V(\theta)-V^*\|_{S_k}^2}{dt} < 0$ when the condition holds. Moreover, the global convergence follows for some sufficiently large $k$ since the effective reversibility coefficient is increasing exponentially with $k$. $\qquad\square$

## D  GENERALIZED DIVERGENCE CONSTRUCTION

**Proposition 1.** *If the MRP is not reversible such that $A = D_\mu(I - \gamma P)$ has at least one non-real eigenvalue, then there exists a function approximator $V$ such that TD learning will diverge. That is, for any initial parameters $\theta_0$, as $t \to \infty$ we have $\|V(\theta) - V^*\| \to \infty$. Moreover, $\nabla V(\theta)$ can have rank up to $n - 1$, where $n$ is the number of states, for all $\theta$.*

*Proof.* Since $A$ is real, having one non-real eigenvalue implies that $A$ has a pair of complex eigenvalues $a + bi, a - bi$ corresponding to eigenvectors $V_1, V_2$. We know that $a > 0$ since $A$ is a non-singular M-matrix and that $b \neq 0$ by assumption. Define $U_1 = Re(V_1)$ and $U_2 = Im(V_1)$ and $U = (U_1 \quad U_2)$. Note that $U_1, U_2$ are linearly independent since $V_1, V_2$ are so that $U$ is full rank and that $AU = U \begin{pmatrix} a & b \\ -b & a \end{pmatrix}$. To construct a divergent approximator, define

$$Q := U(U^T U)^{-1} \begin{pmatrix} -a & b \\ -b & -a \end{pmatrix} (U^T U)^{-1} U^T$$

so that $Q$ is a matrix of rank 2 with range equal to $E := \text{span}\{U_1, U_2\}$.

For any $V \in E$ so that $V = U \begin{pmatrix} x \\ y \end{pmatrix}$ we have $\|V\|^2 \leq C(x^2 + y^2)$ with $C > 0$. Then

$$V^T Q^T A V = (x \quad y) U^T U (U^T U)^{-1} \begin{pmatrix} -a & b \\ -b & -a \end{pmatrix} (U^T U)^{-1} U^T U \begin{pmatrix} a & b \\ -b & a \end{pmatrix} \begin{pmatrix} x \\ y \end{pmatrix}$$

$$= -(x \quad y) \begin{pmatrix} a^2 + b^2 & 0 \\ 0 & a^2 + b^2 \end{pmatrix} \begin{pmatrix} x \\ y \end{pmatrix} \leq -(a^2 + b^2) \frac{\|V\|^2}{C}$$

Thus, $Q^T A$ is a matrix of rank 2 with eigenvectors $V_1, V_2$ and eigenvalues $(a^2 + b^2)i, (a^2 + b^2)i$. Choosing $V_0$ in the span of $V_1, V_2$ we define for $\theta \in \mathbb{R}$:

$$V(\theta) = e^{(Q+\epsilon I)\theta} V_0 \in \text{span}\{V_1, V_2\}$$

Then for $V^* = \mathbf{0}$, we apply the above to (7) to get

$$\dot{\theta} = -\nabla V(\theta)^T A V(\theta) = -V(\theta)^T (\epsilon I + Q^T) A V(\theta) > 0$$

as long as $\epsilon < (a^2 + b^2)/C$ and $V(\theta) \neq \mathbf{0}$. This gives us divergence since as $\theta \to \infty$ we have $\|V(\theta)\| \to \infty$.

Now we just need to show that we can make $\nabla V(\theta)$ rank $n - 1$. To do this we will note that we can define any function $\bar{V}(\bar{\theta})$ from $\mathbb{R}^d \to E^\perp$ for any number of parameters and we still get divergence if we use the function approximator $V'(\theta, \bar{\theta}) = V(\theta) + \bar{V}(\bar{\theta})$. This is because $\frac{\partial V'(\theta,\bar{\theta})}{\partial \theta} = (\epsilon I + Q)V(\theta) \perp A\bar{V}(\bar{\theta})$ so that $\dot{\theta}$ remains strictly above 0 as before, implying divergence. With this construction, we can build a divergent approximator $V'$ on $n$ states with $\nabla V'(\theta, \bar{\theta})$ of rank $n - 1$ everywhere. For example, set $d = n - 2$ and let $\bar{V}(\bar{\theta})$ be the identity function. $\qquad\square$

## E  NEURAL NETWORKS ARE HOMOGENEOUS

**Lemma 1.** *(Liang et al., 2017) Let $\sigma : \mathbb{R} \to \mathbb{R}$ be $h$-homogeneous. Then define a homogeneous network $f : \mathbb{R}^d \to \mathbb{R}^n$ by*

$$f(\theta_1, \cdots, \theta_L) = \sigma(\cdots \sigma(\sigma(\Phi\theta_1)\theta_2) \cdots \theta_L) \tag{25}$$

*where $\theta_i$ is a $d_{i-1} \times d_{i+1}$ matrix and $\Phi$ is the $m \times p$ matrix of data points and $\sigma$ is applied componentwise. Then for all $1 \le i \le L$*

$$f(\theta) = h^{L-i+1} \frac{\partial f(\theta)}{\partial vec(\theta_i)} vec(\theta_i) \qquad and \qquad f(\theta) = \nabla f(\theta) \theta \sum_{i=1}^{L} h^{L-i+1}/L \qquad (26)$$

*where vec maps the matrices $\theta_i$ to vectors while $\theta \in \mathbb{R}^d$ is already a vector.*

*Proof.* We will consider $n = 1$ and note that the result applies for arbitrary $n$ by applying the result to each component function of $f$.

Define $g^i(\theta_1, \ldots, \theta_i) = \sigma(\cdots \sigma(\sigma(\Phi\theta_1)\theta_2) \cdots \theta_{i-1})\theta_i$, which we will denote simply $g^i$ so that we have $f = \sigma(g^L)$. Note that $g^i$ is a $d_{i+1}$ dimensional vector and $d_{L+1} = 1$. Define $f^i(\theta_1, \ldots, \theta_{i-1}) = \sigma(\cdots \sigma(\sigma(\Phi\theta_1)\theta_2) \cdots \theta_{i-1})$, and denote it $f^i$ so that $f^i = \sigma(g^i)$. Let $f^0$ be $\Phi$. Now we proceed by induction on the gap $j - i$ for $j \ge i$ from 1 to $L - 1$.

As the base case, consider $i = j$. Then we have that

$$\frac{\partial f_a^i}{\partial vec(\theta_i)} vec(\theta_i) = \frac{\partial f_a^i}{\partial g^i} \frac{\partial g^i}{\partial vec(\theta_i)} vec(\theta_i) = \sum_e \sum_{b,c} \frac{\partial f_a^i}{\partial g_e^i} \frac{\partial g_e^i}{\partial (\theta_i)_{bc}} (\theta_i)_{bc}$$

$$= \sum_{b,c} \frac{\partial f_a^i}{\partial g_a^i} \frac{\partial g_a^i}{\partial (\theta_i)_{bc}} (\theta_i)_{bc} = \sigma'(g_a^i) \sum_b \frac{\partial g_a^i}{\partial (\theta_i)_{ba}} (\theta_i)_{ba}$$

$$= \sigma'(g_a^i) \sum_b f_b^{i-1} (\theta_i)_{ba} = \sigma'(g_a^i) g_a^i = \frac{1}{h} \sigma(g_a^i) = \frac{1}{h} f_a^i$$

Make the inductive assumption that for $j - i = k \ge 0$ we have for $a \le d_{i+1}$

$$\frac{\partial f_a^j}{\partial vec(\theta_i)} vec(\theta_i) = \frac{1}{h^{k+1}} f_a^j$$

Now consider $j - i = k + 1$. We have

$$\frac{\partial f_a^j}{\partial vec(\theta_i)} vec(\theta_i) = \frac{\partial f_a^j}{\partial f^{j-1}} \frac{\partial f^{j-1}}{\partial vec(\theta_i)} vec(\theta_i) = \frac{1}{h^{k+1}} \sum_b \frac{\partial f_a^j}{\partial f_b^{j-1}} f_b^{j-1}$$

$$= \frac{1}{h^{k+1}} \sum_b \sum_c \frac{\partial f_a^j}{\partial g_c^j} \frac{\partial g_c^j}{\partial f_b^{j-1}} f_b^{j-1} = \frac{1}{h^{k+1}} \sum_b \sigma'(g_a^j)(\theta_j)_{ba} f_b^{j-1}$$

$$= \frac{1}{h^{k+1}} \sigma'(g_a^j) g_a^j = \frac{1}{h^{k+2}} f_a^j$$

Taking $j = L$ gives us the first result that $f(\theta) = h^{L-i+1} \frac{\partial f(\theta)}{\partial vec(\theta_i)} vec(\theta_i)$. Summing over the $L$ choices of $i$ gives $f(\theta) = \nabla f(\theta) \theta \sum_{i=1}^{L} h^{L-i+1}/L$. $\qquad \square$

