# OpenReview forum: "Geometric Insights into the Convergence of Nonlinear TD Learning"
_ICLR.cc/2020/Conference — Accept (Poster)_

### Official Review · AnonReviewer3 · 2019-10-28
**Official Blind Review #3**

**Rating:** 8

**Review:**


####
A. Summarize what the paper claims to do/contribute. Be positive and generous.
####

The paper characterises the convergence of Temporal Difference learning for on-policy value estimation with nonlinear function approximators. It looks at a few classes of functions which include Deep ReLU networks, with and without residual connections. It looks at how a few aspects of function approximators affect their convergence properties, and how these are intertwined with a property of the environment.

This topic is important for improving the theoretical understanding of Deep RL. The paper provides a series of mathematical results which will be interesting for the community as they are well-motivated, intuitively explained, and of clear practical relevance.  The paper works within a simplified setup that is however NOT toy. It makes reasonable simplifying assumptions to avoid confounding issues with exploration, off-policyness of data, and stochastic optimization. Within a continuous-time MRP framework they show that:

1. When a particular class of function approximators known as homogenous functions (e.g. Deep ReLU networks) is used, the error on the state value function can be bounded. Tweaking the class of function approximators by making them like residual networks (residual-homogenous) obtains a bound similar to known bounds for linear function approximators.

2. It is known that TD(0) converges with linear function approximators and arbitrary environments. It is also known that TD(0) converges with arbitrary function approximators when the environment is fully reversible. This paper shows that there is in fact a tradeoff between how well the function approximator is conditioned and how reversible the environment is (for particular definitions of well-conditionedness and the "extent" to which an environment is reversible). This nicely links up known theory and is especially relevant for practitioners who would like to apply neural net function approximators in arbitrary environments.

3. They show that using n-step returns instead of TD(0) returns can have a similar effect to the environment being reversible. That's a nice insight.

4. There is a classic counterexample for TD(0) with a nonlinear function approximator diverging. The paper makes this example more general which more clearly demonstrates how/why convergence fails beyond the single pathological example whose relevance to real neural nets was hard to determine. It makes it clear that a necessary condition for convergent TD learning with function approximators is dodging this more general class of divergent example.

5. The theory is supported by (toy) experiments, whose empirical results suggest the theory can be made more general (e.g. to include more classes of function approximators).

####
B. Clearly state your decision (accept or reject) with one or two key reasons for this choice.
####

This paper should be accepted because the results are interesting, relevant, novel (as far as the reviewer understands), well-explained, and as far as the reviewer can tell correct (though I have not scrutinized the proofs in the appendix).

The paper is interesting and easy to read (even for someone without background in proving the convergence of RL algorithms).

####
C. Provide supporting arguments for the reasons for the decision.
####

Whether or not (and under what conditions) TD(0) converges is an important object of study for the Deep RL community, which is well-represented at ICLR. The results shown here should be more widely known.

Good intuitions given for the mathematical results in addition to proofs (e.g. why homogeneity prevents divergence).

This work contributes to the understanding of Deep RL and could eventually lead to actionable theory which lets us design more robust RL systems (with insights about the coupling between learning algorithm, function approximator, and environment).

####
D. Provide additional feedback with the aim to improve the paper. Make it clear that these points are here to help, and not necessarily part of your decision assessment.
####

D0. My main critique is that the results are somewhat disparate. How does homogeneity or residual homogeneity relate to the conditioning number of the neural tangent kernel? Can these all be connected up better?
D1. Be more clear in Remark 1 on page 4 that the homogenous property applies to *deep* ReLU networks. Otherwise the reader may assume the proof only applies to single-layer neural networks until they read the more detailed exposition in the appendix.
D2. Is it an issue for homogeneity if there is a point where the derivative does not exist (e.g. at x=0 for ReLU).
D3. Can the paper make it more clear to a Deep RL audience in the intro or discussion what the holy grail of this research direction would be?
D4. As tanh or multiplicative activations (e.g. those found in LSTM/attention networks) are not homogenous, can the authors speculate about whether or not they would have similar convergence properties to homogeneous activations? What are the obstacles to a similar proof for this class of networks?.

**Experience Assessment:**

I do not know much about this area.

**Review Assessment: Checking Correctness Of Derivations And Theory:**

I assessed the sensibility of the derivations and theory.

**Review Assessment: Checking Correctness Of Experiments:**

I assessed the sensibility of the experiments.

**Review Assessment: Thoroughness In Paper Reading:**

I read the paper thoroughly.

---

> ### Author Response · Authors · 2019-11-11
> **response**
>
> We thank the reviewer for a very thorough review.
>
> Here are responses to the feedback provided in section D of the review:
>
> D0. We agree that connecting the results more tightly is a worthwhile goal. We began from the two known regimes (linear functions and reversible environments) and in attempting to close the gap were able to extend results from both sides, but not in such a way as to unify the two settings. Reviewer 1 raised a similar point and we will add a comment to the text to clarify this as a future direction.
>
> D1. We will add this clarification.
>
> D2. Thank you for the comment, this raises an interesting technicality. While the ReLU is not differentiable at zero, it does not change the essence of the results since at zero the non-differentiability is masked in the equation x*sigma’(x) = sigma(x) (also in practice sigma’(0) is usually defined to be 0). Some care needs to be taken to make sure that the dynamics are defined at the non-differentiable points, but since our analysis considers only the pointwise sign of the Lyapunov function we can use the masking observation so that all the algebraic manipulations will remain the same. Also note that a similar issue was already addressed by Chizat and Bach in https://arxiv.org/abs/1805.09545 by reparametrizing the class of ReLU functions.
>
> D3. This is an interesting question. In our mind the dream result in this direction would be an understanding of these semi-gradient algorithms in the same level of detail as gradient descent in supervised learning. For example, in that case we have a precise notion of the assumptions needed to get global convergence (i.e. convexity), guarantees of convergence and characterizations of the fixed points in hard non-convex problems (local minima), and even convergence rates under various assumptions. We can add a comment about this to the discussion to motivate further work in this area.
>
> D4. As we explained in the paper, homogeneity is nice since then the manifold of functions has the property that the function V(theta) lies in the span of the tangent space nabla V(theta) at that point. It is not as easy to say things about this manifold for non-homogeneous activations. There may be other ways to think about the benefit of homogeneity that generalize to neural nets with different activations, but we are not familiar with any.

---

### Official Review · AnonReviewer2 · 2019-11-01
**Official Blind Review #2**

**Rating:** 6

**Review:**

The paper analyses the convergence of TD-learning in a simplified setup (on-policy, infinitesimally small steps leading to an ODE).

Several new results are proposed:
- convergence of TD-learning for a new class of approximators (the h-homogenous functions)
- convergence of TD-learning for residual-homegenous functions and a bound on the distance form optimum
- a relaxation of the Markov chain reversibility to a reversibility coefficient and convergence proof that relates the reversibility coefficient to the conditioning number of grad_V grad_V^T.

While the theory applies to the ideal case, t provides some practical conclusions:
- TD learning with k-step returns  converges better because the resulting Markov chain is more reversible
- convergence can be attained by overparmeterized function approximators, which can still generalize better than tabular value functions.

The experiments corroborate the link between reversibility factor and TD convergence on an artificial example.

**Experience Assessment:**

I do not know much about this area.

**Review Assessment: Checking Correctness Of Derivations And Theory:**

I assessed the sensibility of the derivations and theory.

**Review Assessment: Checking Correctness Of Experiments:**

I assessed the sensibility of the experiments.

**Review Assessment: Thoroughness In Paper Reading:**

I read the paper at least twice and used my best judgement in assessing the paper.

---

> ### Author Response · Authors · 2019-11-11
> **response**
>
> We thank the reviewer for the positive review and are happy to provide any clarifications that could convince the reviewer to increase the score.

---

### Official Review · AnonReviewer4 · 2019-11-04
**Official Blind Review #4**

**Rating:** 3

**Review:**

This paper establishes a theoretical insight into Temporal Difference (TD) learning for policy evaluation, on the convergence issue with nonlinear function approximation. It proposes that for a so-defined class of “homogeneous” function approximators, the TD learning will be attracted to a “neighborhood” of the optimal solution. While this seems an important work, I am uncomfortable to give an accept decision because the statements of the results can be found inaccurate from place to place. I actually found it is a bit confusing to use this wording (from the paper). For example, with neural networks, there is still approximation error and local minima issue, how could you say that the update is absorbed into a neighborhood of the true solution? And this is claimed in the beginning of Section 3.

TD learning follows a biased estimate of the gradient of the squared expected Bellman error, which
is minimized by the true value function. The bias is intrinsic to the fact that one cannot obtain more
than one independent sample from the environment at any given time. As it turns out, this bias
can be seen geometrically as “bending” the gradient flow dynamics and potentially eliminates the
convergence guarantees of gradient descent when combined with nonlinear function approximation.
》》 I don’t know what this means. TD diverges with nonlinear FA just because the contraction mapping does not hold any more.

such as two timescale algorithms, but these algorithms are not widely used
>>this argument is a bit weak.

We prove global convergence to the true value function when the environment is “more
reversible” than the function approximator is “poorly conditioned”.
>>not clear what this means until here. What is “reversible environment”, what does it mean FA is “poorly conditioned”? Later in Section 2, it was mentioned “MRP” is reversible so that some matrix is symmetric.

Section 2:

Equation 1 uses V^* is a bit inconsistent ( P is used). Why not use V? V^* usually means the optimal value function. I saw your footnote, but remember value function is “associated” with P.

Convergence to V* immediately follows. – What convergence? I thought you were talking about stability of the ODE.

the “semi-gradient” TD(0): do you mean tabular TD(0) is not semi-gradient? Do you think it is gradient? Even in tabular, it is not gradient descent.

V(\theta)_s: this notation is odd.

it is meant to approximate gradient descent on the squared expected Bellman error:  This is arguable. Actually it is not precise. One can say it is true and others may say it’s not. This is never an established result or acknowledge showing that TD is an approximation to the gradient descent on the mean squared Bellman error.


The first is when V is linear and the second when the MRP is reversible so that A is symmetric.
>>this is ambiguous. Do you mean the second case is when V is nonlinear and the MRP is reversible? I am guessing this is what you mean. And it’s true.

Section 2.3: doesn’t carry much value. The example is from the paper cited (Tsitsiklis and Vanroy 1997). Adding the symmetric case for P doesn’t give much value because that’s easily seen to be true.

Definition 1: “homogeneous”. This is not the usual definition of homogeneous in mathematics. Square function is.  Relu: gradient at 0 exist?

 Section 4.2: experiments about modifying the spiral example into symmetric MRP is not very interesting, because symmetry brings obvious convergence guarantee. However, it is good to see the experiment with a variable delta that controls the level of symmetry.

I think a missing experiment is the showcase for the divergence examples the case of “homogeneous” function, such as the square function and the neural networks (as claimed in the paper, these are “homogenous” functions). How does the behavior that the TD update is absorbed into the “neighborhood” of the true value function?









**Experience Assessment:**

I have published in this field for several years.

**Review Assessment: Checking Correctness Of Derivations And Theory:**

I assessed the sensibility of the derivations and theory.

**Review Assessment: Checking Correctness Of Experiments:**

I carefully checked the experiments.

**Review Assessment: Thoroughness In Paper Reading:**

I read the paper at least twice and used my best judgement in assessing the paper.

---

> ### Author Response · Authors · 2019-11-11
> **response**
>
> We thank the reviewer for the detailed comments and we will respond to each of the comments in order.
>
> By neighborhood we mean a set containing the true value function. In this case the set that the approximate value function is attracted to is a ball in the mu-norm with radius B as defined in Theorem 1, which  contains the true value function.
>
> Here we were referring to how sometimes TD is motivated as an approximation to a gradient descent algorithm and in fact is gradient descent in reversible environments. This comparison to gradient descent is a main theme of the paper. We agree that the way this was explained in the sentence in question was confusing though and we will change it in the paper. And to clarify, just because the contraction mapping does not hold uniformly anymore once we add approximation, this does not necessarily imply divergence.
>
> Our point here is not that two timescale algorithms are inherently bad (this would require much stronger evidence), but rather that it is an interesting problem to analyze TD(0) directly. We can remove this clause to prevent confusion.
>
> We did not want to put all of the definitions into the introduction, so rather put a high level description of the results to come later. We can clarify that here reversibility refers to the MRP in the usual sense of Markov chain reversibility.
>
> As we said in the footnote, in our setting there is only one policy and thus only one value function to find so we used V^*. V is already used to represent the map from parameters to functions and we did not want to overload notation even more. We also went back and forth on this, would it be better to use V^pi instead of V^* to show that it is the value function of whatever policy induces the transitions?
>
> We agree this is a little imprecise, we mean that the flows defined by this ODE will converge to a unique equilibrium point regardless of initial conditions. We will make the appropriate update.
>
> We agree that tabular TD is also gradient descent. We are referring to how the algorithm is defined in Sutton and Barto, as referenced in that sentence. In that book (which is a common reference) the method is called semi-gradient to emphasize that the  and to differentiate it from the gradient TD (GTD) algorithm.
>
> We agree that the V(theta) notation is non-standard, but our analysis relies on considering the dynamics in the space of function and so we need this mapping from parameters to functions. Tsitsiklis and Van Roy among others use a similar notation.
>
> We agree that TD is not GD and maybe not even motivated by GD of the squared expected bellman error. Again we are referring to the introduction of the algorithm and discussion in Sutton and Barto.
>
> Yes, we mean when the environment is reversible irrespective of the linearity of the function (it could be linear or nonlinear).
>
> Section 2.3 is meant to provide intuition and motivate the results to come. It is not presented as a new result (it is in the “Setup” section of the paper). We think it provides a useful visualization, especially for those less familiar with older papers like Tsitsiklis and Van Roy.
>
> We agree that this is confusing. The definition is equivalent to the traditional definition of positive homogeneity. See for example: https://en.wikipedia.org/wiki/Homogeneous_function#Positive_homogeneity.  We will clarify this in the paper. Regarding the gradient at 0, reviewer 1 made a similar point, see our D2 in our response to them.
>
> We agree that these experiments are not so surprising, but the point is just to verify some of the intuitions from the paper not to provide novel empirical results.
>
> We agree that adding some experimental confirmation of the homogeneity results would be nice, but we are not sure exactly how to provide such evidence. Empirically, we have not seen any non-convergent behavior with homogeneous functions in any toy environments we have constructed. But simply showing some examples where the behavior is good could just mean that we were choosing easy problems. Would it be useful to show that in the same or similar environment as the other experiment we get convergence with simple homogeneous functions?
>
>
>
> We hope that these changes provide the necessary clarification and ask the reviewer to let us know if anything remains unclear.

---

### Official Review · AnonReviewer1 · 2019-11-04
**Official Blind Review #1**

**Rating:** 8

**Review:**

What is the specific question/problem tackled by the paper?

The paper studies convergence & non-divergence of TD(0) with value function estimates from the class of ReLU deep neural nets (optionally with residual connections).

Is the approach well motivated, including being well-placed in the literature?

Yes.


Does the paper support the claims? This includes determining if results, whether theoretical or empirical, are correct and if they are scientifically rigorous.

The support is adequate.


Summarize what the paper claims to do/contribute. Be positive and generous.

The paper takes a first step in bridging the gap between existing analyses of TD(0): convergence with linear function approximators, and convergence in reversible MRPs. The first contribution is a non-divergence result for the method with ReLU deep neural networks with and without residual connection. The second result connects a notion of reversibility of an MRP and the condition number of the neural tangent kernel, saying that better conditioning can make up for the lack of reversibility.


Clearly state your decision (accept or reject) with one or two key reasons for this choice.

I vote for accepting the paper.


Provide supporting arguments for the reasons for the decision.

The paper is well written, with a clear story and accessible explanations. The paper provides novel results and an interesting line of work that allows us to tradeoff good behavior of the function space and of the MRP in order to have TD converge, in the sense that as the MDP becomes less and less reversible we can make up for it by having properly conditioned matrices.

The paper also makes an adequate choice of function space to restrict the guarantees to.


Provide additional feedback with the aim to improve the paper. Make it clear that these points are here to help, and not necessarily part of your decision assessment.

It seems that Theorem 3 cannot recover the linear case results for properly conditioned Phi matrices and irreversible MRPs. It would be good if the result could really interpolate between the two previously studied cases (linear irreversible and nonlinear reversible). Alternatively, a comment about this limitation of the result would improve the paper.

**Experience Assessment:**

I have published one or two papers in this area.

**Review Assessment: Checking Correctness Of Derivations And Theory:**

I assessed the sensibility of the derivations and theory.

**Review Assessment: Checking Correctness Of Experiments:**

I assessed the sensibility of the experiments.

**Review Assessment: Thoroughness In Paper Reading:**

I read the paper thoroughly.

---

> ### Author Response · Authors · 2019-11-11
> **response**
>
> We thank the reviewer for the careful review.
>
> In regards to Theorem 3, the reviewer makes a good point that the Theorem does not recover the results from the linear case. We think that closing this gap is an interesting direction and will add a comment about this limitation to the discussion section of the paper (the homogeneous case gives an extension of the linear setting but not in a way that closes this gap).

---

### Decision · Program_Chairs · 2019-12-19

**Decision:**

Accept (Poster)

**Comment:**

This paper takes steps towards a theory of convergence for TD(0) with non-linear function approximation.  The paper provides two theoretical results.  One result bounds the error when training the sum of linear and homogenous parameterized functions.  The second result shows global convergence when the environment dynamics are sufficiently reversible  and the differentiable function approximation is sufficiently well-conditioned.  The paper provides additional insight using a family of environments with partially reversible dynamics.

The reviewers commented on several aspects of this work.  The reviewers wrote that the presentation was clear and that the topic was relevant.  The reviewers were satisfied with the correctness of the results.  The reviewers liked the result that state value function estimation error is bounded when using homogeneous functions. They also noted that the deep networks in common use are not homogeneous so this result does not apply directly. The result showing global convergence of TD(0) with partial reversibility was also appreciated. Finally, the reviewers liked the family of examples.

This paper is acceptable for publication as the presentation was clear, the results are solid, and the research direction could lead to additional insights.